# Dynamics of Growth in Purebred Pacu (*Piaractus mesopotamicus*) and Tambaqui (*Colossoma macropomum*), and Their Reciprocal Hybrids, under Varied Feeding Programs: Insights from Nonlinear Models

**DOI:** 10.3390/genes14101976

**Published:** 2023-10-23

**Authors:** Woshinghton Rocha Gervaz, Antônio Fernando Leonardo, Diogo Teruo Hashimoto, Ivan Bezerra Allaman, Gabriel Rinaldi Lattanzi, Rafael Vilhena Reis Neto

**Affiliations:** 1UNESP Aquaculture Center (CAUNESP), Via de Acesso Prof. Paulo Donato Castellane, Jaboticabal 14884-900, Brazil; w.gervaz@unesp.br (W.R.G.); diogo.hashimoto@unesp.br (D.T.H.); gabrielunesp2015@hotmail.com (G.R.L.); 2São Paulo Agribusiness Technology Agency (APTA), Regional Polo of Vale do Ribeira, Rodovia BR 116, km 460, São Paulo 11900-000, Brazil; antonio.leonardo@sp.gov.br; 3Department of Exact and Technological Sciences, State University of Santa Cruz, Jorge Amado Highway, km 16, Bairro Salobrinho, Ilhéus-Bahia 45662-900, Brazil; ivanalaman@gmail.com; 4UNESP Faculty of Agricultural Sciences of Vale do Ribeira—Campus of Registro, Avenida Nelson Brihi Badur 430, Registro 11900-000, Brazil

**Keywords:** growth potential, growth rate, hybridization, serrasalmidae

## Abstract

We evaluated the growth performance of pacu and tambaqui and their reciprocal hybrids (tambacu and paqui) under different feeding programs. We raised 30 individuals from each species and their respective crosses, distributing them into three replicate cages with 10 individuals each. Throughout the 5-month experimental period, the fish were weighed six times while exposed to diverse feeding regimens involving three commercial diets with varying combinations of crude protein (CP) levels: 24%, 28%, and 32%. Growth curves were adjusted using nonlinear models. The evaluation period was insufficient for adjusting the logistic model for the tambaqui and paqui treatments with the highest initial protein intake. Pure pacu had a higher (*p* < 0.05) growth rate (k = 0.0185) than in the tambacu hybrid (k = 0.0134) and proved to have an early performance since animals from this group reached their weight at inflection faster. Despite growing more slowly, tambacu reached a higher (*p* < 0.05) final weight (A = 1137.12) than in the pacu (A = 889.12). Among the feeding programs, animals that received less CP showed greater growth potential; however, longer evaluation is important to verify if the initial CP intake has no significant positive effect on fish growth.

## 1. Introduction

Within the last 10 years, Latin America has substantially increased its contribution to world aquaculture, accounting for 18% of total fish production in 2020. Brazil emerged as the eighth largest producer of fish in continental waters globally in 2020, with a remarkable output of 552 thousand tons [1]. Based on national statistical data, Brazilian aquaculture achieved a notable production of 860 thousand tons of fish in 2022, with tilapia (*Oreochromis* sp.), an exotic species, accounting for 64% of the total output [2].

However, Brazil has other native fish species and hybrids for national aquaculture that should be studied as economically interesting options for fish farmers. The production of species from the Serrasalmidae family, such as pacu (*Piaractus mesopotamicus*), tambaqui (*C. macropomum*), and their reciprocal crosses, has a prominent position in Brazilian aquaculture. Collectively, their production constitutes 38% of the total Brazilian aquaculture output, with an estimated yield of 288 tons in 2019 [3].

Neotropical fish from the Serrasalmidae family are not only popularly raised in Brazil but also widespread across numerous countries in South and Central America, as well as several countries in the Caribbean and Asia, including China, Indonesia, Malaysia, Myanmar, and Vietnam [4]. In China, total production of tambaqui, pacu, pirapitinga (*Piaractus brachypomus*), tambatinga (*C. macropomum* × *P. brachypomus*), and tambacu (*C. macropomum × P. mesopotamicus*) amounts to approximately 575 tons per annum [5].

In regions with colder temperatures during autumn and winter, fish farmers undertake the crossbreeding of tambaqui and pacu species to amalgamate two crucial traits: the enhanced growth potential of tambaqui and the cold tolerance exhibited by pacu [6]. However, although this strategy is widely used, studies that prove the advantage of producing hybrid fish over “pure” parental species under different farming conditions remain scant [7]. Moreover, the absence of viable methods for fish stock identification and effective management of breeding programs leads to erroneous mating, resulting in the production of backcrossed animals with lower performance than in the F1 hybrids.

Multiple authors have drawn attention to the environmental concerns arising from the utilization of specific potentially fertile hybrids and non-native species in aquaculture production [8,9,10,11,12,13]. Non-native species accidentally escaping from aquaculture can genetically interact with autochthonous wild fish populations, posing a threat to their survival [10,14].

Crossbreeding is an animal breeding strategy widely used in production species, such as cattle, swine, and poultry, and has been widely applied in aquaculture, eventually benefiting fish farmers. However, given the context already discussed, it is important to scientifically verify the advantages of farming hybrids before encouraging them, since evidence indicates that the hybrids produced by crossing pacu and tambaqui are potentially fertile [13].

Although using interspecific hybrids in aquaculture offers potential benefits to fish farmers, scientific verification of the hybrids’ potential and optimizing their production technology before introducing them into commercial aquaculture is essential. Feeding management is fundamental to optimizing animal production, which involves selecting the ideal feed type and feeding strategy. Commercial diets are typically formulated by adjusting the crude protein content based on the specific feeding habits of the fish species in question. Commercially, no specific formulations are available for the hybrids, and these animals may respond differently to the nutritional requirements of their parental species [7].

An informative way of evaluating animal performance is using the growth curve, a graphical representation of weight as a function of the individual’s age. Ref. [15] demonstrated that studying growth by fitting functions that describe an animal’s lifespan allows for the consolidation of information from a series of data into a concise set of interpretable biological parameters. Therefore, it becomes feasible to identify populations or groups of animals that attain their consumption size at a younger age by utilizing growth curve parameters. According to [16], this valuable information can be obtained by examining the growth curve parameters K and A.

In growth curve modeling, the K parameter represents the asymptotic or maximum size an animal can reach. In contrast, parameter A refers to the age at which the growth rate of the animal is highest. By analyzing these parameters, it can gain essential information for understanding the growth patterns of different populations or groups of animals.

In this study, we investigated the weight–age relationship of two neotropical fish species from the Serrasalmidae family, namely pacu (*P. mesopotamicus*) and tambaqui (*C. macropomum*), and their reciprocal hybrids, tambacu and paqui, under varying feeding regimes. We employed nonlinear models to analyze their growth patterns, exploring the impact of different feeding programs on their development.

## 2. Methodology

### 2.1. Animals and Experimental Conditions

The experiment was conducted in Pariquera-açu (24°43′14″ S, 47°52′43″ W) at the Aquaculture Station of the Regional Pole of the Agribusiness Technology Agency of São Paulo, APTA Regional, Brazil.

A diallel cross was conducted between pacu (5 males and 5 females) and tambaqui (5 males and 4 females) using the artificial reproduction protocol described by [17], resulting in the generation of four experimental fish groups: (1) pure pacu (female pacu × male pacu), (2) pure tambaqui (female tambaqui × male tambaqui), (3) hybrid tambacu (female tambaqui × male pacu), and (4) hybrid paqui (female pacu × male tambaqui). Before conducting the crosses, the animals were evaluated using the multiplex-PCR technique of nuclear (tpm1 and rag2) and mitochondrial (mt-co1 and mt-cyb) genes to ensure they originated from purebred parents. Further details regarding the reproduction and evaluation of the parents, which were utilized to create the genetic groups, have been comprehensively described in [7].

A total of thirty 9-month-old fish from each group were utilized for a 5-month experimental rearing period, spanning from November 2016 to March 2017, under distinct feeding conditions. At the onset of the experiment, the average body mass of fish from each group was as follows: tambaqui = 186 ± 22.3 g, pacu = 141 ± 18.6 g, tambacu = 128 ± 16.3 g, and paqui = 154 ± 18.9 g. The fish were individually microchipped, and then ten individuals from each group cohabitated within three cages of 4.8 m^3^ each (2 m × 2 m × 1.2 m). The cages were placed in 600 m^2^ flow-through excavated ponds with an average depth of 1.5 m, where the average water exchange rate was 25% per day. The cages were positioned at a minimum distance of 50 cm from the bottom of the pond to maintain optimal sanitary conditions.

Program 1 (P1) involved single-phase feeding, utilizing a diet containing 24% crude protein throughout the experimental rearing period. Feeding program 2 (P2) comprised two phases: Phase 1 (1st month) and Phase 2 (2nd to 5th month), wherein the fish received diets with 28% and 24% crude protein (CP), respectively. Feeding program 3 (P3) consisted of three phases: Phase 1 (1st month), Phase 2 (2nd month), and Phase 3 (3–5th month), when the fish received diets with 32%, 28%, and 24% CP, respectively.

All commercial diets were purchased from the same manufacturer, and their respective nutritional compositions were guaranteed by the manufacturer (TrowNutrition^®^, Campinas, Brazil) on the product label (Table 1). The energy value of the diets was determined by adding the calories contributed by proteins, lipids, and non-nitrogen extract (ENN) multiplied by the Atwater factors [18]. ENN was derived by subtracting the total weight of the moisture, protein, ether extract, and mineral matter. The animals were fed twice daily, at 9:00 a.m. and 3:30 p.m., until apparent satiety. Additionally, the cages were monitored daily for any signs of dead fish.

Water parameters were measured weekly between the two lines of the cages using a multiparameter water analyzer (HI9146-04: Hanna Instruments, Barueri, Brazil). Throughout the experiment, the mean (±standard deviation) of dissolved oxygen values at depths of 10 cm, 70 cm, and 150 cm were 7.9 ± 1.9 mg/L, 7.7 ± 1.7 mg/L, and 6.2 ± 1.7 mg/L, respectively. At the corresponding depths, the pH values were 6.15 ± 1.1, 5.9 ± 0.9, and 6.0 ± 0.7, respectively. The water temperature, measured at a depth of 50 cm, ranged from 25.2 °C to 29.8 °C, with an average of 27.4 ± 3.8 °C. All recorded parameters were within the ranges recommended by CONAMA (Resolution 357/05) for farming aquatic organisms in continental waters [19].

All animals were weighed six times throughout the experimental period at 1, 30, 67, 96, 128, and 158 days of the study. Before weighing, the fish were anesthetized in a tank containing water and eugenol at a concentration of 100 mg/L [20]. Subsequently, they were identified and weighed using a precision balance. Average daily weight gain (*DWG*) was calculated for each individual using the following equation [21]:$$DWG\left(g\right)=\frac{\left(Final\;weight-Initial\;weight\right)}{Experiment\;days}.$$

All animal handling procedures followed the established standards set forth by the Ethics Committee on Animal Use of the Faculty of Agrarian and Veterinary Sciences of UNESP (CEUA, 12 April 2015, Protocol 23291/15).

### 2.2. Statistical Analysis

The data obtained from the final weighing, along with the average daily weight gain (*DWG*), were analyzed using a hierarchical model that accounted for the effects of the mating system (pure breed or crossbreed) and the experimental groups as follows: tambaqui and pacu nested in the “pure breed” category, and tambacu and paqui nested in the “crossbreed” category. For these analyses, the nutritional plan’s effect was incorporated as a block in the model, and the initial weight of the fish was included as a covariate.

Heterosis was determined for final weight and DWG based on the averages adjusted by initial body mass, using the following formula:$$\;H\left(\%\right)=\frac{\left[\frac{\left(PQ+TC\right)}{2}\right]-\left[\frac{\left(PC+TQ\right)}{2}\right]}{\frac{\left(PC+TQ\right)}{2}}\times 100$$
where *H* represents heterosis, $\frac{\left(PQ+TC\right)}{2}$ is the average of hybrids (*PQ* = paqui group; *TC* = tambacu group), and $\frac{\left(PC+TQ\right)}{2}$ is the average of purebreds (*PC* = pacu group; *TQ* = tambaqui group).

Compiled data from all weighings were analyzed in a mixed linear model with the fixed effects of food programs, experimental groups, and age in a 3 × 4 × 6 factorial scheme (nutritional plans × experimental groups × age), and, as a strategy to isolate individual effects within the experimental group, such as behavioral dominance, we included the random effect of each fish in the model.

Tukey’s test was employed for multiple comparisons of the means when significant differences were observed between different groups or food programs. Additionally, when a significant interaction effect was detected between the age factor and experimental groups or food programs, a logistic model (1) was utilized to fit the growth curves, following the reparameterization (2) outlined below:(1)$$y={A\left(1+Be^{-kt}\right)\;}^{-1}$$
(2)$$y=\frac{A}{1+B^{-k*t\;}}.$$

Parameter “*A*” represents the maximum or ultimate weight the fish can reach as it matures, measuring the upper limit of its growth in the model. Parameter “K” represents the growth rate of the animal as it approaches its asymptotic weight “*A*”. A higher value of “K” indicates that the animal reaches its adult weight more quickly, indicating faster growth (younger fish), while a lower value suggests a slower growth rate. Parameter “B” is a constant in the logistic growth equation, helping in the mathematical representation of the growth curve. Finally, parameter “*t*” represents the time (days) variable in the logistic growth equation and signifies the duration over which the growth is observed.

When the logistic model did not fit the data set, the exponential model was used:$$y=Ae^{kt}$$

In this case, “*A*” represents the estimated initial weight, “K” the relative growth rate, and “*t*” is time.

The logistic and exponential models’ parameters were assessed for the feeding programs and experimental groups using the likelihood ratio test. To ensure the validity of our results, we thoroughly evaluated all assumptions, and in the cases where violations occurred, we applied logarithmic transformations. The chosen significance level for the statistical tests was 5%.

The entire data analysis was conducted using R software version 4.2.3 [22], a widely recognized and powerful tool for statistical analysis.

## 3. Results

No mortality was observed during the experimental period. The hierarchical model showed the superiority (*p* < 0.05) of the pure groups concerning the hybrids for final weight and daily growth rate (DWG), resulting in negative heterosis for both traits (Table 2). Regarding the hierarchical model, a difference between the pure groups was observed only for the daily weight gain, with pacu (3.5 ± 0.98 g) presenting a higher average (*p* < 0.05) than in the tambaqui (3.12 ± 1.1 g). Among the hybrid groups, no significant differences (*p* > 0.05) were observed in any trait.

The mixed model revealed a statistically significant effect (*p* < 0.05) for both the feeding programs and experimental groups. However, the interaction between these factors was not significantly different (*p* > 0.05), indicating that the feeding programs for pure and hybrid fish may yield similar outcomes (Table 3).

The post-hoc comparison test, conducted on the average weight adjusted using the mixed model, indicated a significant difference (*p* < 0.05) between pure and hybrid animals. However, no significant difference (*p* > 0.05) was observed among the animals within each pure group (pacu × tambaqui) or between the animals in the two hybrid groups (tambacu × paqui). Regarding the feeding programs, the test of means revealed a statistically significant superiority (*p* < 0.05) in the weight of fish fed with Program 1 (Figure 1).

Considering the significant interactions uncovered using the mixed model (Table 3), we adjusted weight versus age curves for each experimental fish group and feeding program. The data for the pure pacu and tambacu hybrid were fitted to a logistic model, revealing a deceleration in the growth of these fishes toward the end of the experiment (Figure 2).

The likelihood test applied to the parameters of the logistic model showed that pacu had a higher relative growth rate (parameter K = 0.0185) (*p* < 0.05) than in the tambacu (k = 0.0134) (Table 4), justifying the higher weight and daily weight gain in this group (Table 2). On the contrary, the asymptotic or estimated maximum weight (Parameter A) was observed to be significantly higher (*p* < 0.05) for the tambacu group (1137.12 g) in comparison to pacu (889.12 g). This observation suggests that despite the slower growth, the hybrid group has greater potential for achieving a higher final weight than in the pure group (Table 4 and Figure 2).

The exponential model could better explain the growth of the pure tambaqui and hybrid paqui groups, suggesting that until the end of the experimental period, the fish in those groups had accelerated growth (Figure 3).

The exponential model has only two parameters: one describing the growth rate (Parameter K), which corresponds to the same parameter as the logistic model, and the other describing the estimated initial weight (Parameter A). The K parameters of the paqui (0.0091) and tambaqui (0.0083) groups were similar (*p* > 0.05). In contrast, the A parameter of the tambaqui group (168.28 g) was higher (*p* < 0.05) than that of the paqui group (141.53 g) (Table 5 and Figure 3). For these two groups, it can be suggested that the estimated initial weight explains the better final performance of the pure group than that of the hybrid (Table 2).

Regarding the feeding programs, we applied the logistic model to adjust the weight data of fish that were fed under Programs 1 and 2, as illustrated in Figure 4.

Throughout the experimental period, fish nourished with the same commercial diet (P1) had a significantly higher growth velocity (K = 0.0159) and greater growth potential (A = 1214.08) compared to animals exposed to the diet stratified by age (P2) (K = 0.0131; A = 1137.42), implying that providing a higher protein diet (28% CP) during the initial month of the experiment did not yield any notable growth benefits (Table 6 and Figure 4).

The exponential model best described the growth of animals subjected to Program 3, which described an estimated growth rate (Parameter K) and initial weight (Parameter A) of 0.0082 and 144.3, respectively (Table 7). This finding holds significant implications, as suggested by the exponential model, which indicates that the fish continued to exhibit rapid growth even at the end of the experiment. Therefore, the stratification of diets based on the age of the fish under Program 3 may imply a positive growth response for the evaluated experimental fish groups.

## 4. Discussion

To date, research on the growth performances of pure and hybrid fishes from the Serrasalmidae family has been conducted, yet the outcomes have been inconsistent. Our findings align with those of [7], who reported a significantly higher final body weight in pure pacu fish (WB = 847.1 g) compared to the tambacu hybrids (WB = 735.1 g). In contrast, [21] reported a significant superiority in the final weight of the tambacu hybrid (1273.06 g) compared to pure pacu species (907.6 g). [23], while evaluating the diallel cross between pacu, tambaqui, and pirapitinga (*P. brachypomus*) species, observed the highest general combining ability for the final weight in the pure tambaqui group.

Despite these discrepancies, our research contributes to the body of knowledge in this field and sheds further light on the growth dynamics of these fish species. An essential difference between the cited studies is the farm system in which the animals were evaluated, which may account for the disparity in their outcomes. Refs. [21,23] conducted their experiments in ponds and recirculation systems, respectively, while [7], similar to this study, evaluated the animals in cages. This difference in the farm system could potentially explain the observed similarities in results between [7] and our study.

Considering other significant factors, such as the evaluation period, is crucial when interpreting the final weight results. In the studies by [21,23], the animals reached a stage close to market weight (1500 g). However, in the study conducted by [7], the final period of growth for the animals, where the growth potential of each genetic group could be determined, was not evaluated. This emphasizes the importance of studying a growth curve that estimates the growth potential and the speed of animal growth. By considering both factors, a more comprehensive understanding of the animals’ growth patterns can be achieved.

In their study, [24] employed a logistics model to compare the growth curves of pacu and tambacu. The authors reported that the hybrid tambacu, exhibited superiority with a higher asymptotic weight (Parameter A) and had the same relative growth rate (Parameter K) as pacu. Moreover, after data adjustment, the pure tambaqui group showed a much higher growth potential (A = 1056.82 g) than in the paqui hybrid (A = 797.82 g). In contrast, the growth rate was similar between groups (K = 0.034 for tambaqui and K = 0.033 for paqui). Based on these results, the authors indicated a better growth performance of tambaqui than in the paqui.

As listed in Table 3, the tambacu hybrid group displayed a higher asymptotic weight than in the pacu, indicating a greater growth potential. However, regarding the relative growth rate, the pure pacu group outperformed the hybrid group, indicating that pacu showed more rapid early growth when evaluated in cages. Ref. [24] evaluated the animals in ponds, which could account for the contrasting relative growth rate findings observed in our study when the animals were evaluated in cages. This highlights the significance of considering the environmental context when interpreting growth performance results.

In practical terms, the parameter “K” of the logistic curve demonstrates how quickly the animals reach adult or asymptotic weight [25]; therefore, in the evaluation conditions, the pacu group would be interesting for a fish industry that aims to process smaller animals. However, if the industry prefers to process larger fish, pacu is no longer a preferred option since, based on the asymptotic weight described by Parameter A of the logistic model (889.12 g), the animals of this group have decelerated growth, which reduces performance for weight gain. However, the tambacu hybrid group is a stronger alternative if the objective is to process larger fish since the fish in this group have a higher asymptotic weight (1137.12 g); the growth of tambacu remains accelerated until this weight.

The data collected for the two other genetic groups, tambaqui and pacu, presented a different scenario, as the adjusted growth model followed an exponential pattern (Figure 3). Consequently, as was carried out for the other groups, drawing direct inferences regarding slaughter and processing weight was not feasible. To assess the growth potential of these groups accurately, it is essential to conduct evaluations over a longer period of time. This extended observation would provide valuable insights into the growth dynamics of tambaqui and pacu, allowing for a comprehensive understanding of their growth potential.

As previously mentioned, [23] arrived at a conclusion affirming the genetic superiority of the tambaqui group over the other three genetic groups (pacu, tambacu, and paqui). This finding was further supported by [26] in their study, where tambaqui outperformed both tambacu and another hybrid (Tambatinga) produced from species within the Serrasalmidae family (*C. macropomum × P. brachypomus*). In this study, we conducted a Tukey test on the averages generated using the mixed model, and the results demonstrated the superior performance of tambaqui compared to both paqui and tambacu (Figure 1). These findings reinforce the advantage of tambaqui in terms of growth and performance, further solidifying its prominence among the evaluated genetic groups.

While the hierarchical model utilized in this study does not permit direct comparisons between groups with different mating systems (pure vs. cross), the results listed in Table 2 reveal significant differences in the final body weight of tambaqui (WBM = 656.7 g), which was 10.2% higher than that of paqui (WBM = 589.6 g) and 16.8% higher than that of tambacu (WBM = 545.9 g). These findings highlight notable variations in growth outcomes among the different experimental fish groups, further emphasizing the distinct growth potential of tambaqui compared to paqui and tambacu.

Fish subjected to Feeding Programs 1 and 2 attained a final weight sufficient for estimating the parameters of the logistic model. Surprisingly, Program 1 yielded fish with higher growth potential (A = 1214.08 g), a faster growth rate (K = 0.0159), and superior performance compared to Program 2 (A = 1137.42 g; K = 0.0131) (Table 4 and Table 5). Despite the expectation, supported by [7], that the higher initial protein intake in Program 2 would positively affect animal growth, as they observed a positive effect on growth for tambacu hybrids due to increased protein supply in the early stages of life, our study’s results contradicted this assumption. Similar to our research, the mentioned researchers obtained comparable results concerning the pure pacu fish.

Several hypotheses can be proposed to explain the results of our study on food programs. Firstly, the 30-day period during which the animals were subjected to Food Program 2, with a higher amount of CP, might not have been long enough to positively affect their growth. Alternatively, it is possible that reducing the protein intake for fish after 30 days of the experiment could have negatively impacted growth, sparing those animals that began the experiment with a lower CP intake from this adverse effect. Furthermore, there could be variations in diet quality, despite being sourced from the same manufacturer. However, none of these hypotheses can be confirmed with the data obtained in this experiment.

The fish fed under Food Program 3 did not attain a final weight sufficient to adjust the logistic growth model (Figure 4), thereby preventing the estimation of their growth potential, which might be higher or lower than that of the fish subjected to feeding Programs 1 and 2. The Tukey test revealed superior performance for fish-fed Program 1 (Figure 2); however, a more extended evaluation of the experiment could potentially unveil a positive growth response in individuals from Program 3, considering their higher CP supply during the initial months of the experiment, similar to what was observed by [7] in their evaluation of the same feeding programs, where superior performance was noted for fish-fed programs with higher protein intake at the beginning of the experiment.

## 5. Conclusions

Our study’s results emphasize the vital importance of studying growth curves, as a superficial analysis based solely on the final performance of the fish without considering specific objectives can potentially lead to incorrect recommendations for the selection of pacu and tambaqui in aquaculture production. Thus, the growth curves provided in our study offer relevant information enabling the appropriate selection of the examined fish groups, aligning with the interests of the aquaculture production industry. Thus, pacu proved to be a precocious species under the evaluation conditions, as the animals in this group reached their weight at inflection more quickly than that of tambacu in cages. In contrast, despite growing slower, the tambacu hybrid could achieve a higher final weight without decelerating its growth. This valuable information is relevant to fish farmers and the fish industry when deciding between these two genetic groups.

In addition to comparing the purebred species of pacu (*P. mesopotamicus*) and tambaqui (*C. macropomum*), along with their reciprocal hybrids (tambacu and paqui), the growth curves obtained proved to be valuable for assessing feeding programs. Animals that received less protein throughout the experiment demonstrated greater growth potential among the feeding programs. While a preliminary analysis indicates that Program 1 was more effective, resulting in better performing animals, a more comprehensive evaluation is necessary before entirely disregarding Program 3. Thus, further investigation is required to make an informed decision about the suitability of Program 3 for achieving optimal growth results. Moreover, conducting a more extended evaluation is crucial to verifying whether the initial intake of this nutrient has no actual positive effect on fish growth.

## Figures and Tables

**Figure 1 genes-14-01976-f001:**
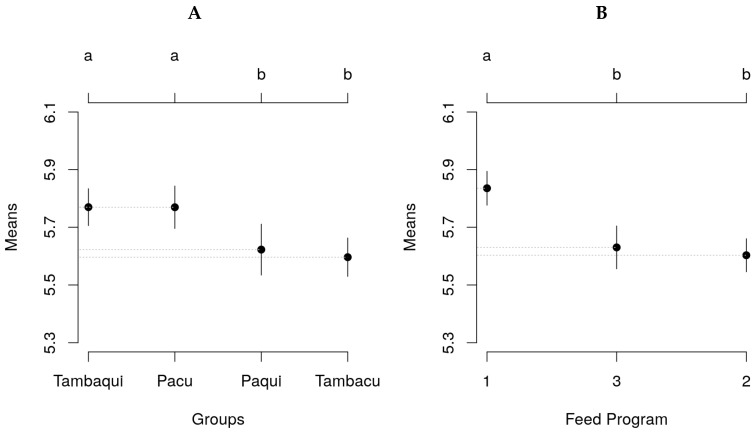
Significant mean differences (Tukey’s test) in weight between experimental fish groups (**A**) and food programs (**B**) with their respective 95% confidence intervals. Lowercase letters indicate statistical differences between treatments.

**Figure 2 genes-14-01976-f002:**
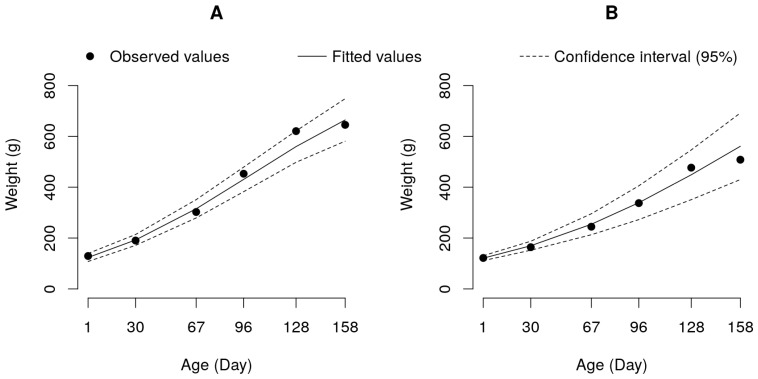
Logistic growth curve adjusted to data on weight by age for the pure pacu (**A**) and tambacu hybrid (**B**) fish groups.

**Figure 3 genes-14-01976-f003:**
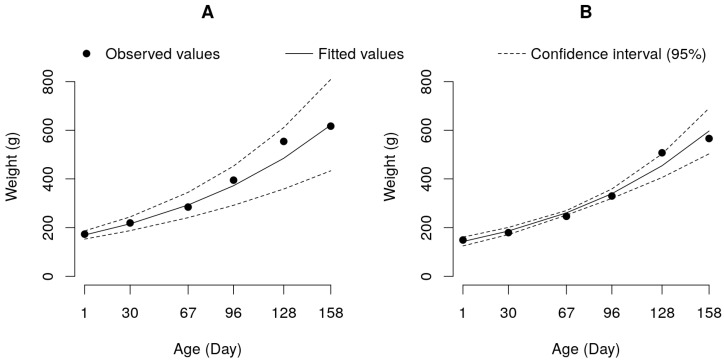
Exponential growth curve adjusted to data on weight by age for the pure tambaqui (**A**) and paqui hybrid (**B**) groups.

**Figure 4 genes-14-01976-f004:**
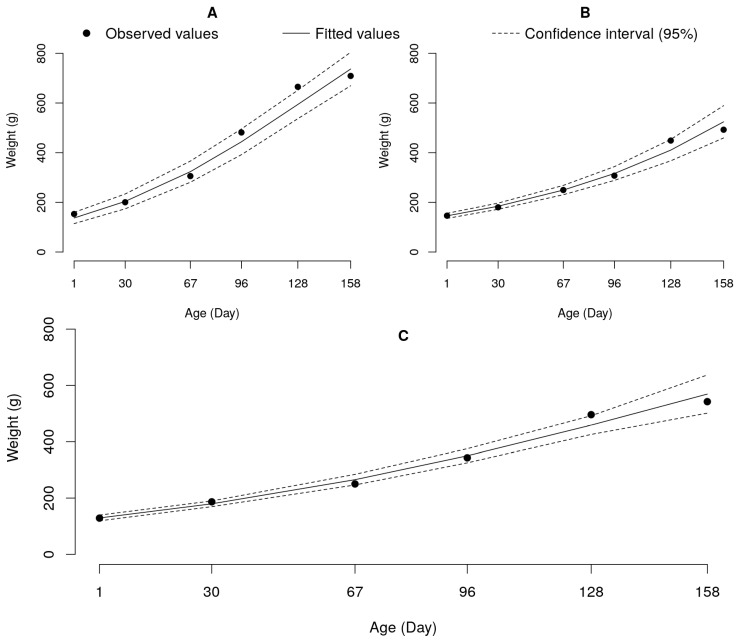
Logistic growth curves adjusted to weights as functions of age for animals fed with food Programs 1 (**A**) and 2 (**B**), and exponential for animals fed with Program 3 (**C**).

**Table 1 genes-14-01976-t001:** Nutritional composition of commercial diets used in the experiment.

	Commercial Diets		
	24% CP *	28% CP **	32% CP ***
Moisture (g/kg)	120	120	120
Crude protein (g/kg)	240	280	320
Lipids (g/kg)	30	35	45
Mineral matter (g/kg)	120	125	120
Crude fiber (g/kg)	120	120	50
Non-nitrogen extract (g/kg)	370	320	345
Calcium (g/kg)	10	10	10
Phosphorus (mg/kg)	6080	6080	6080
Energy (MJ/kg)	13.30	13.57	15.35
Protein: Energy	18.1	20.6	20.8
Carbohydrate: Lipid	12.3	9.1	7.6

* Diet 24% CP: Na 2800 mg/kg, Mg 25 mg/kg, Zn 72 mg/kg, Cu 16.02 mg/kg, Fe 28.02 mg/kg, Mn 40.08 mg/kg, Co 0.4808 mg/kg, I 1.044 mg/kg, Se 0.2088 mg/kg, Folic acid 3.408 mg/kg, Niacin 72 mg/kg, Biotin 0.36 mg/kg, Pantothenic acid 22.44 mg/kg, Vit B1 12.84 mg/kg, Vit B12 14.4 μg/kg, Vit B2 12.84 mg/kg, Vit B6 12.84 mg/kg, Vit C 150 mg/kg, Vit D3 2001.6 IU/kg, Vit E 88.2 IU/kg, Vit K3 6.42 mg/kg, Vit A 5923.2 IU/kg, Inositol 48.06 mg/kg. ** Diet 28% CP and *** Diet 32% CP, respectively: Na 2500, 1300 mg/kg; Mg 25 mg/kg; Lysine 11, 14.4 g/kg; Methionine 4300 mg/kg; Zn 72, 96 mg/kg; Cu 16.02, 21.96 mg/kg; Fe 28.02, 37.36 mg/kg; Mn 40.08, 53.44 mg/kg; Co 0.4808, 0.5344 mg/kg; I 1.044, 1.3920 mg/kg; Se 0.2088, 0.2784 mg/kg; Folic acid 3.408, 5.544 mg/kg; Niacin 72, 96 mg/kg; Biotin 0.36, 0.48 mg/kg; Pantothenic acid 22.44, 29.92 mg/kg; Vit B1 12.84, 17.12 mg/kg; Vit B12 14.4, 19.2 μg/kg; Vit B2 12.84, 17.12 mg/kg; Vit B6 12.84, 17.12 mg/kg; Vit C 200, 250 mg/kg; Vit D3 2001.6, 2668.8 IU/kg; Vit E 88.2, 117.6 IU/kg; Vit K3 6.42, 8.53 mg/kg; Vit A 5923.2, 7897.6 IU/kg; Inositol 48.06, 64.08 mg/kg; Colin 348 mg/kg (only for 32% CP diet).

**Table 2 genes-14-01976-t002:** Mean values of the final body weight, average daily weight gain, and heterosis for performance variables of purebred (pacu and tambaqui) and hybrid (tambacu and paqui) neotropical fish experimentally farmed for this study.

Variables	Purebreds		Purebred Means	Hybrids		Hybrid Means	Heterosis (%)	C.V. (%)
	Pacu	Tambaqui		Tambacu	Paqui			
Final W (g)	667.7^(170)^	656.7^(192)^	662.2^(181)^ A	545.9^(158)^	589.6^(145)^	567.75^(151)^ B	−14.3	17.1
DWG (g)	3.5^(0.98)^ a	3.12^(1.1)^ b	3.31^(1)^ A	2.81^(0.95)^	2.89^(0.81)^	2.85 ^(0.88)^ B	−13.9	29.8

Mean values of purebred and hybrid animals followed by different capital letters and the means of experimental groups within purebred or hybrid animals followed by different lowercase letters significantly differ according to the Scott–Knott test (*p* < 0.05). Final W = final weight, DWG = average daily weight gain, C.V. = coefficient of variation.

**Table 3 genes-14-01976-t003:** Mixed model analysis of variance for the logarithmic weight variable of neotropical fish farmed over 5 months in cages under this study.

Component	Sum Sq	MeanSq	NumDF	F Value	Pr (>F)
Genetic Groups	0.152	0.051	3	4.891	0.0041
Food Programs	0.262	0.13	2	12.615	0.0000
Age	98.861	19.772	5	1902.964	0.0000
Group * Programs	0.108	0.018	6	1.729	0.1294
Group * Age	1.61	0.107	15	10.331	0.0000
Program * Age	1.91	0.191	10	18.383	0.0000
Group * Program * Age	0.436	0.014	30	1.399	0.0852

* means interaction between factors, Sum Sq = sum of squares, MeanSq = mean squares, NumDF = degrees of freedom.

**Table 4 genes-14-01976-t004:** Parameters of the logistic model estimated for the pacu (pure) and tambacu (hybrid) groups with their respective confidence intervals (95%).

Parameters	Pacu			Tambacu		
	Inferior Limit	Estimates	Upper Limit	Inferior Limit	Estimates	Upper Limit
A	653.7	889.12	1124.67	407.68	1137.12 *	1866.57
B	4.66	6.31	7.96	2.63	8.51	14.39
K	0.0153	0.0185 *	0.0218	0.0095	0.0134	0.0172

A = asymptotic or estimated final weight, B = model constant, K = relative growth rate. * Different estimates between genetic groups by the likelihood test (5%).

**Table 5 genes-14-01976-t005:** Parameters of the exponential model estimated for the tambaqui (pure) and paqui (hybrid) groups with their respective confidence intervals (95%).

Parameters	Tambaqui			Paqui		
	Inferior Limit	Estimates	Upper Limit	Inferior Limit	Estimates	Upper Limit
A	151.76	168.28 *	184.8	123.4	141.53	159.65
K	0.0068	0.0083	0.0097	0.0074	0.0091	0.0109

A = estimated initial weight, K = relative growth rate. * Different estimates between genetic groups by the likelihood test (5%).

**Table 6 genes-14-01976-t006:** Parameters of the logistic model estimated for animals with different feeding programs and their respective confidence intervals (95%).

Parameters	Feed Program 1			Feed Program 2		
	Inferior Limit	Estimates	Upper Limit	Inferior Limit	Estimates	Upper Limit
A	830.56	1214.08 *	1597.6	170.58	1137.42	2104.26
B	5.1	7.99	10.89	0.6010	7.89	15.17
K	0.0133	0.0159 *	0.0185	0.0091	0.0131	0.0171

A = asymptotic or estimated final weight, B = model constant, K = relative growth rate. * Different estimates between genetic groups by the likelihood test (5%).

**Table 7 genes-14-01976-t007:** Parameters of the exponential model estimated for animals fed Program 3 and their respective confidence intervals (95%).

Parameters	Inferior Limit	Estimates	Upper Limit
A	134.2	144.34	154.48
K	0.0074	0.0082	0.0089

A = estimated initial weight, K = relative growth rate.

## Data Availability

Data will be provided on request.

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
