# Peer review of "Dynamics of Growth in Purebred Pacu (Piaractus mesopotamicus) and Tambaqui (Colossoma macropomum), and Their Reciprocal Hybrids, under Varied Feeding Programs: Insights from Nonlinear Models"

_genes, 2023, doi:10.3390/genes14101976_

Round 1

Reviewer 1 Report

Rafael et al have investigated the growth of fish fed with different feed programs. I would suggest the authors to consider some points discussed below and revise the manuscript accordingly.

Please include line numbers next time so it will be easy for reviewers to comment specific to the line.

In abstract, “different feeding was used for each cage”. It is each group that received different feeding program or each cage received different feeding program.

The authors applied statistical analysis to investigate the growth of pure and hybrid breds. But only the growth is enough to substantiate the hypothesis. Eventhough the authors mentioned that shorter evaluation did not have more informative parameters.

The figures and tables show the obtained results. The authors have to check the figure panel alphabets in figure 1.

Minor language editing is required.

Reviewer 2 Report

Dear Editors,

Dear Authors,

The manuscript titled "Nonlinear Models Explain the Growth of Pure and Hybrid Neotropical Fish Fed with Different Food Programs" presents an intriguing and valuable study. The research aims to elucidate the growth performance of pacu (Piaractus mesopotamicus) and tambaqui (Colossoma macropomum), along with their reciprocal hybrids, tambacu and paqui, under various feeding programs. These species hold significant importance for aquaculture production in South America and Asia. The findings offer valuable insights that can enhance the efficiency of aquaculture production for these fish. Moreover, the manuscript has the potential for publication and can contribute significantly our understanding of the growth dynamics in neotropical fish species.

However, it is essential to acknowledge that the overall quality of information presentation, especially in terms of language and logical organization, requires substantial improvement. Therefore, I do not recommend the manuscript for publication in the present form and major revision should be made by Authors. The attached file contains detailed remarks pointing out areas that need enhancement.  All remarks, questions and fixes were placed in the attached pdf file (yellow highlights contain fixes and sentence suggestions, while red highlights contain comments and questions).

Dear Editors,

Dear Authors,

The manuscript titled "Nonlinear Models Explain the Growth of Pure and Hybrid Neotropical Fish Fed with Different Food Programs" presents an intriguing and valuable study. The research aims to elucidate the growth performance of pacu (Piaractus mesopotamicus) and tambaqui (Colossoma macropomum), along with their reciprocal hybrids, tambacu and paqui, under various feeding programs. These species hold significant importance for aquaculture production in South America and Asia. The findings offer valuable insights that can enhance the efficiency of aquaculture production for these fish. Moreover, the manuscript has the potential for publication and can contribute significantly our understanding of the growth dynamics in neotropical fish species.

However, it is essential to acknowledge that the overall quality of information presentation, especially in terms of language and logical organization, requires substantial improvement. Therefore, I do not recommend the manuscript for publication in the present form and major revision should be made by Authors. The attached file contains detailed remarks pointing out areas that need enhancement.  All remarks, questions and fixes were placed in the attached pdf file (yellow highlights contain fixes and sentence suggestions, while red highlights contain comments and questions).

Thank you for another interesting manuscript that I could review!

Round 2

Reviewer 2 Report

Dear Editors,

Dear Authors,

The manuscript entitled: “Dynamics of Growth in Purebred Pacu (Piaractus mesopotamicus) and Tambaqui (Colossoma macropomum), and their Reciprocal Hybrids, under Varied Feeding Programs: Insights from Nonlinear Models” has been significantly improved. The Authors have regarded all my remarks and suggestions. The manuscript represents good quality study. I do not have any other remarks and the manuscript can be published in the present form. Nice work!